# Genetic Analysis of Early White Quality Protein Maize Inbreds and Derived Hybrids under Low-Nitrogen and Combined Drought and Heat Stress Environments

**DOI:** 10.3390/plants10122596

**Published:** 2021-11-26

**Authors:** Olatunde A. Bhadmus, Baffour Badu-Apraku, Oyenike A. Adeyemo, Adebayo L. Ogunkanmi

**Affiliations:** 1Department of Cell Biology and Genetics, University of Lagos, Lagos 101017, Nigeria; azeezolatunde@yahoo.com (O.A.B.); aoadeyemo@unilag.edu.ng (O.A.A.); adebayoogunkanmi@yahoo.com (A.L.O.); 2International Institute of Tropical Agriculture, IITA, PMB 5320 Oyo Road, Ibadan 200001, Nigeria

**Keywords:** QPM inbred lines, combined drought and heat stress, maize, low soil nitrogen, drought

## Abstract

An increase in the average global temperature and drought is anticipated in sub-Saharan Africa (SSA) as a result of climate change. Therefore, early white quality protein maize (QPM) hybrids with tolerance to combined drought and heat stress (CDHS) as well as low soil nitrogen (low-nitrogen) have the potential to mitigate the adverse effects of climate change. Ninety-six early QPM hybrids and four checks were evaluated in Nigeria for two years under CDHS, low-nitrogen, and in optimal environments. The objectives of this study were to determine the gene action conditioning grain yield, assess the performance of the early QPM inbred lines and identify high yielding and stable QPM hybrids under CDHS, low-nitrogen and optimal environment conditions. There was preponderance of the non-additive gene action over the additive in the inheritance of grain yield under CDHS environment conditions, while additive gene action was more important for grain yield in a low-nitrogen environment. TZEQI 6 was confirmed as an inbred tester under low N while TZEQI 113 × TZEQI 6 was identified as a single-cross tester under low-nitrogen environments. Plant and ear aspects were the primary contributors to grain yield under CDHS and low-nitrogen environments. TZEQI 6 × TZEQI 228 and the check TZEQI 39 × TZEQI 44 were the highest yielding under each stress environment and across environments. Hybrid TZEQI 210 × TZEQI 188 was the most stable across environments and should be tested on-farm and commercialized in SSA.

## 1. Introduction

Maize (*Zea mays* L.) is an economically important annual cereal crop that is predicted to become the most important cereal crop in the world by 2025 [1]. The current global yield of maize is about 1.1 billion tons and production is projected to increase to over 1.4 billion tons by 2030 [1,2]. It is estimated that 50% of the total population of West and Central Africa (WCA) depends on maize as a staple food while a large proportion of the maize produced annually is used as raw materials for various alcoholic beverages, poultry and the livestock industries [3,4]. As a staple food crop of the sub-region, maize plays an important role in combating malnutrition. The normal endosperm maize has about 71% starch compared to the quality protein maize (QPM) which has 46% lysine and 66% tryptophan in addition to the 71% starch. The two amino acids supply provide about 70–73% of the requirement of the human body [4,5,6]. Early studies by Akuamoah- Boateng [7] on infants fed with QPM porridge revealed reduced growth stunting, fewer sick days, and healthier growth statistics than those fed with normal endosperm maize. Other studies by Mbuya et al. [8] and Panda et al. [9] reported that QPM could replace soybean in poultry feed production which could reduce the cost of animal feed.

In addition to the agro-ecological advantage, the savannas of SSA contribute to high maize productivity through high incoming solar radiation and reduced incidence of pests and diseases due to prevailing low humidity and low night temperature conditions [4,10]. However, tropical soils have low levels of organic matter and available nitrogen, resulting in nutrient-depleted soils which cause grain yield losses of up to 80% [11,12]. Thus, resource-poor farmers constituting a large proportion of the maize producers in SSA can hardly afford mineral fertilizers due to the high cost of nitrogen-based fertilizers as well as the general non-availability of fertilizers [4,13]. Furthermore, climate change threatens the goal of achieving global food security and could have severe socio-economic consequences globally [14]. With the fast-increasing world population, maize production is expected to be significantly affected by the adverse impacts of climate change and could lead to a global food crisis with major impacts, particularly in SSA [15,16,17,18]. Although maize is well adapted and substantially utilized in the savannas of SSA, the concurrent incidence of abiotic stresses such as drought and high temperature during flowering could reduce the photosynthetic rate, accelerate leaf senescence, induce kernel abortion and ultimately cause drastic yield losses [19,20]. The combination of the two stresses could lead to a grain yield loss of more than 90% during flowering and grain filling in maize [10,21,22,23,24]. 

According to Neate [25], 40% of the maize production areas in SSA will be unsuitable for cultivation of the available maize germplasm due to the threat posed by CDHS as well as lack of tolerance to low-nitrogen by 2030. A major challenge of maize breeders of the present generation is to develop maize cultivars with CDHS and low-nitrogen tolerance for the agro-ecological zones of SSA [4,15]. A number of early and extra-early maturing maize QPM hybrids with tolerance to *Striga*, low-nitrogen and drought stress have been developed by the International Institute of Tropical Agriculture Maize Improvement Program (IITA-MIP) and commercialized in West and Central Africa (WCA) [4,26]. However, very little information is available on QPM maize hybrids with tolerance to CDHS. While several studies have been conducted on the improvement of maize varieties for tolerance to drought and heat stress separately, Cairns et al. [27] highlighted that tolerance to both stresses separately did not confer tolerance to CDHS and concluded that drought and heat stresses in maize were genetically different from tolerance to the individual stresses. This calls for the need for the development and commercialization of hybrids with CDHS and low-nitrogen tolerance for SSA to mitigate the adverse effects of these stresses on farmers, particularly the resource-poor farmers.

Maize breeding efforts in recent years have recorded significant successes in identifying lines that could be used as tolerance donors for breeding of cultivars with CDHS tolerance. Cairns et al. [15] identified donor lines among CIMMYT and IITA inbred lines possessing high levels of tolerance to CDHS that could be used for the development of outstanding hybrids. In another study, Meseka et al. [21] assessed the level of tolerance in existing drought tolerant hybrids under CDHS and reported an appreciable level of tolerance among the hybrids. Similarly, Nelimor et al. [22] assessed early maize landraces from Ghana, Burkina-Faso and Togo, and identified accessions that could be invaluable donors of tolerant alleles for hybrid development under CDHS conditions. However, very few reports are available on the genetic action regulating grain yield and other agronomic traits under CDHS.

Breeding of early white QPM maize hybrids with tolerance to CDHS and the information on the gene action involved in the inheritance, the combining ability and heterotic patterns of inbred lines in the IITA-MIP are crucial to their successful use for hybrid development and production [26,28]. A recent study by Chiuta and Mutengwa [24] reported the preponderance of the non-additive gene action in the inheritance of grain yield of subtropical QPM inbreds under CDHS suggesting that hybrid development could be adopted as an efficient and effective breeding strategy for the development of tolerant cultivars [26]. Contrarily, Nasser et al. [29] studied the combining ability of early maturing yellow maize inbreds and reported the importance of the additive gene action over the non-additive gene action in the inheritance of grain yield under CDHS. The inconsistencies in the reports on the relative importance of the additive and non-additive gene action in the inheritance of grain yield and other traits call for the need for further studies to confirm the type of gene action controlling grain yield and other agronomic traits in the newly developed early maize QPM inbred lines in the IITA-MIP. Additionally, several studies have reported the preponderance of additive gene action over non-additive gene action in the inheritance of grain yield under low-nitrogen conditions [12,30,31,32]. Contrarily, studies by Makumbi et al. [33] and Badu-Apraku et al. [26] reported that non-additive gene action conditioned the grain yield of maize hybrids under low-nitrogen, while Noelle et al. [34] reported that both additive and non-additive genetic action conditioned grain yield when soil nitrogen was low. These conflicting reports by researchers could be due to the severity of the low-nitrogen environments or the germplasm used in the studies. Consequently, there is a need to evaluate the gene action conditioning the inheritance of grain yield and other important traits under low-nitrogen environments using the newly developed early white QPM inbred lines of the IITA-MIP. 

The availability and use of early white QPM hybrids with tolerance to CDHS and low-nitrogen conditions would be more sustainable for smallholder farmers in nitrogen-depleted areas of SSA where low levels of agricultural inputs are utilized and there is simultaneous incidence of drought and heat stresses [4]. Therefore, the objectives of this study were to: (a) investigate the gene action conditioning grain yield and other agronomic traits of early white QPM inbred lines under CDHS, low-nitrogen, as well as optimal growing environments; (b) group the inbred lines into heterotic groups using the heterotic grouping method based on the combining ability of multiple traits (HGCAMT) and identify inbred and single-cross testers; (c) assess the performance of the QPM inbred lines in hybrid combinations and identify high yielding and stable QPM hybrids under CDHS, low-nitrogen and across test environments.

## 2. Results

### 2.1. Analysis of Variance of Agronomic Traits

Under the CDHS environments, the analysis of variance (ANOVA) revealed significant (*p* < 0.01) variations among environment (E) and genotype (G) mean squares for the measured traits except for E mean squares for plant height and G mean squares for anthesis-silking interval (ASI) and plant height. Significant differences for genotype by environment interaction (GEI) mean squares were also observed for measured traits except for plant and ear heights (Table 1). Partitioning the hybrid components of variation into male(set) (GCAm), female(set) (GCAf) and female × male(set) (SCA) mean squares, GCAm and GCAf revealed significant differences in mean squares for most measured traits except GCAm effect for grain yield, ASI, plant height, plant and ear aspects as well as GCAf effect for ASI, plant height, ear aspect and tassel blasting. Additionally, significant SCA effects were observed for grain yield, days to 50% anthesis, ear height and tassel blasting. Significant differences were observed for GCAf and SCA and in their interaction with E for grain yield, days to 50% silking and the stay green characteristic. Broad sense heritability estimates ranged from 26% for the stay green characteristic to 74% for ear height while grain yield recorded 37%. In general, low to high H^2^ estimates were observed for the measured traits. Moderately high repeatability estimates were recorded for measured traits under CDHS.

Across the low-nitrogen environments, the ANOVA revealed significant (*P* < 0.01) differences in mean squares for E, G and GEI for grain yield and other measured traits except G mean squares for ASI and plant aspect and GEI mean squares for ASI (Table 1). Significant differences were observed for GCAm and GCAf mean squares for measured traits except GCAm mean squares for ear height, ear aspect and the stay green characteristic as well as GCAf mean squares for ASI and ear height. No significant SCA mean squares were observed for measured traits. Additionally, significant differences in GCAf mean squares and the interactions with the environment were observed for grain yield, plant, and ear aspects. Broad sense heritability estimates ranged from 12% for ear aspect to 52% for plant height. Grain yield had H^2^ estimate of 37%. Generally, very low H^2^ estimates were observed for the measured traits. Repeatability estimates of the measured traits were generally high for all locations. Consequently, the data for all locations were included in the analysis of variance.

Under the optimal environment conditions, significant (*p* < 0.01) differences in mean squares were observed for E, G and GEI for the measured traits, except the G and GEI mean squares for grain yield and ear rot (Table 2). In addition, significant differences (*p* < 0.01) were observed for GCAm, GCAf and SCA mean squares for measured traits except GCAf and SCA mean squares for grain yield and ear rot. Furthermore, significant mean squares for GCAm × E, GCAf × E and SCA × E were observed for measured traits except for grain yield. Broad-sense heritability estimates varied from 13% for ears per plant to 78% for plant height while grain yield recorded H^2^ of 41%. Moderate to high repeatability estimates were recorded for measured traits. Across research environments, significant variations (*p* < 0.01) were observed among E, G and GEI mean squares for measured traits except the GEI for plant height (Table 2). Significant differences were observed for GCAm, GCAf, and SCA mean squares except the SCA mean squares for grain yield, days to 50% silking, plant height and ear rot. Significant mean squares for GCAm × E, GCAf × E and SCA × E were also observed for measured traits except for plant height and GCAm × E mean squares for plant aspect, plant height and ear rot as well as SCA × E mean squares for grain yield, plant aspect and plant height (Table 2).

### 2.2. Proportionate Contributions of Genetic Variances under Low-Nitrogen, CDHS, Optimal and across Test Environments

The percentage contributions of GCA and SCA variances were calculated as the proportion of the GCA components to the total genetic variance based on the sum of squares [35]. The percentage contribution of GCA (GCAm and GCAf) to the overall genotypic sum of squares for grain yield was higher under low-nitrogen (60.1%) than under CDHS (43.1%), optimal conditions (43.2%) and across stress conditions (53.3%). However, the percentage contribution of SCA sum of squares was highest under CDHS (56.9%) and optimal conditions (56.7%, Figure 1, Table 3). Under low-nitrogen environments, the contributions of GCA sum of squares (GCAm and GCAf) were predominantly higher than those of the SCA sum of squares for measured traits ranging from (46.6%) for anthesis-silking interval to (76.7%) for plant height. Generally, the contribution of the GCA sum of squares was greater than the SCA sum of squares for grain yield and the other agronomic traits except for ASI. Under CDHS environment, the contribution of SCA sum of squares to the overall genotypic sum of squares among hybrids was greater than the GCA sum of squares for grain yield (56.9%), ASI (60.4%), plant height (50.3%), ear aspect (61.8%), ears per plant (67.7%) and the stay green characteristic (52.9%) (Figure 1 and Table 3). Under optimal conditions, the contribution of the SCA sum of squares were greater than GCA sum of squares for grain yield, ASI, ear aspect and ears per plant. Additionally, the contribution of the GCA sum of squares ranged from 32.3% for ears per plant to 62.1% for ear height. Across test environments, the contribution of GCA sum of squares was higher than the SCA sum of squares for most measured traits. The proportion of the GCA sum of squares relative to the total sum of squares of the hybrids varied from 38.5% for ASI to 77.1% for days to 50% silking while the contribution of the SCA sum of squares ranged from 22.8% for days to 50% silking to 61.5% for ASI across test environments. The GCA sum of squares accounted for 51% of the total sum of squares attributable to grain yield (Table 3).

### 2.3. Estimates of General Combining Ability Effects of the 24 QPM Inbred Lines 

Under low nitrogen environment conditions, the GCAm effects for grain yield ranged from -318.88 for TZEQI 165 to 567.43 for TZEQI 158 while the GCAf effects ranged from -612.63 for TZEQI 240 to 961.61 for TZEQI 210 (Table 4). Only TZEQI 6 had significant and positive GCAm and GCAf effects for grain yield. Inbreds TZEQI 106, TZEQI 113, TZEQI 158, TZEQI 188, and TZEQI 6 had significant and positive GCAm effects for grain yield while TZEQI 210 and TZEQI 6 had significant and positive GCAf effects for grain yield. Also, only TZEQI 106 recorded significant and negative GCAm and GCAf effects for the stay green characteristic. TZEQI 130, TZEQI 140, TZEQI 216 and TZEQI 6 had significant and negative GCAm effects for the stay green characteristic while TZEQI 210 had significant and negative GCAf effects for the stay green characteristic. Under CDHS conditions, the GCAm effects for grain yield varied from -633 for TZEQI 219 to 503 for TZEQI 216. No significant GCAf effects were recorded for any of the inbred lines, however TZEQI 113 and TZEQI 216 recorded significant and positive GCAm effects for grain yield. Inbred lines TZEQI 113, TZEQI 130, TZEQI 171 and TZEQI 188 had significant and negative GCAm effects for ASI while TZEQI 171, TZEQI 210, TZEQI 241and TZEQI 6 recorded significant and negative GCAf effects for ASI. Of the 24 parental lines, only TZEQI 219 had significant and positive GCAm effects for grain yield under optimal environments (Table 4). Across test environments, GCAm effects for grain yield varied from -388 for TZEQI 241 to 685 for TZEQI 219 while the GCAf effects varied from -530 for TZEQI 240 to 452 for TZEQI 210. Significant and positive GCAm effects were recorded for TZEQI 219 and TZEQI 6 while significant and positive GCAf effects were detected for TZEQI 210 and TZEQI 241.

### 2.4. Heterotic Grouping of Inbred Lines Based on General Combining Ability of Multiple Traits (HGCAMT) Method

Across test environments, the dendrogram constructed using the HGCAMT method revealed four groups when 40% of the variation was explained (r^2^ = 0.4) among the parental lines (Figure 2). Eleven inbred lines (TZEQI 106, TZEQI 210, TZEQI 241, TZEQI 123, TZEQI 171, TZEQI 162, TZEQI 130, TZEQI 143, TZEQI 175, TZEQI 176, TZEQI 216) were placed in heterotic group I while TZEQI 132, TZEQI 188, TZEQI 219, TZEQI 159, and TZEQI 240 constituted heterotic group II. Inbreds TZEQI 113, TZEQI 158, TZEQI 6 and TZEQI 140 were placed in heterotic group III with the remaining four inbred lines TZEQI 122, TZEQI 246, TZEQI 228 and TZEQI 165 classified into heterotic group IV. Inbred lines TZEQI 113, TZEQI 158, and TZEQI 6 with significant and positive GCA effects for grain yield were placed in the same heterotic group III. Interestingly, inbred lines TZEQI 106 and TZEQI 210 with significant and positive GCA effects for grain yield under low-nitrogen environments were placed in heterotic group I.

### 2.5. Identification of Inbred and Single-Cross Hybrid Testers

Based on the criteria proposed by Pswarayi and Vivek [36], inbred TZEQI 6 which had high per se grain yield as well as significant and positive GCA effects for grain yield was identified as an inbred tester under low-nitrogen conditions. Hybrid TZEQI 113 × TZEQI 6 was also identified as a single-cross tester.

### 2.6. Grain Yield in Contrasting Environments

Under CDHS environments, grain yield ranged from 342 kg ha^−1^ for TZEQI 143 × TZEQI 122 to 3885 kg ha^−1^ for hybrid check TZEQI 39 × TZEQI 44 with a mean of 1578 kg ha^−1^. The hybrid check out-yielded the highest yielding QPM hybrid by 7% (Table 5). Grain yield of the QPM hybrids under low-nitrogen environment, ranged from 1511 kg ha^−1^ for TZEQI 132 × TZEQI 122 to 5388 kg ha^−1^ for TZEQI 210 × TZEQI 188 with a mean of 3057 kg ha^−1^. TZEQI 210 × TZEQI 188 out-yielded the highest yielding hybrid check by 18%. Under optimal conditions, grain yield ranged from 3807 kg ha^−1^ for TZEQI 241 × TZEQI 216 to 6692 kg ha^−1^ for TZEQI 210 × TZEQI 188 with a mean of 5164 kg ha^−1^. Comparison of the grain yield of the QPM hybrids under CDHS environment to that of optimal environments showed a wide reduction ingrain yield (5–93%) with a mean of 67% while grain yield reduction under low-nitrogen environments revealed a yield reduction range of 8% to 74% with a mean of 44.9%. Overall, the CDHS and low-nitrogen tolerant QPM hybrids recorded lower reductions in grain yield than the susceptible QPM hybrids (Table 5).

### 2.7. Grain Yield Stability of Hybrids across Test Environments

The observed significant differences for genotype by environment interactions for grain yield under CDHS, low-nitrogen and across test environments justified the need to investigate the grain yield stability of the early QPM hybrids across the test environments using the genotype plus genotype by environment (GGE) biplot procedure. The GGE biplot analysis revealed that the first and second principal component axes explained 65.5% of the total variation in grain yield of the QPM hybrids (Figure 3 and Figure 4). The “which-won-where” view of the GGE biplot procedure was used to identify hybrids adapted to specific environments. In the polygon view, the vertex entries represented the highest yielding hybrids in the environments that fell within the sector. The distance between the hybrids and the biplot origin measured the differences in the performance of the hybrids and how they differed from the mean yields of other hybrids. Therefore, the vertex hybrids in each sector were more responsive to the environments than those that fell within the polygon or located close to the biplot origin. There were eight vectors with hybrids 1, 2, 6, 10, 17, 20, 24, and 25 as the vertex hybrids (Figure 3). Two environments E2 and E6 fell within the sector where hybrids TZEQI 241 × TZEQI 216 and TZEQI 6 × TZEQI 55 (check) were the vertex hybrids. Similarly, environments E1 and E3 fell within the sector where hybrids TZEQI 6 × TZEQI 210 and TZEQI 39 × TZEQI 44 (check) were the vertex hybrids and were therefore considered the highest yielding in those environments. Environments E4, E5 and E7 had no vertex hybrids and therefore no outstanding hybrids were identified in those environments. Entries 1, 2, 6, and 10 were vertex hybrids but they were not identified with any environments. Entries 13 and 15 (TZEQI 171 × TZEQI 158 and TZEQI 210 × TZEQI 171) were the least responsive hybrids to environmental variability due to their closeness to the biplot origin. 

The “mean vs. stability” of the GGE biplot view was used to identify the highest yielding and the most stable QPM hybrids across the test environments (Figure 4). The vertical line that passed through the biplot origin (intercept of the vertical and horizontal axis) is referred to as the average tester coordinate (ATC). The ATC ordinate separated the low yielding hybrids from the high yielding hybrids. The average yields of the QPM hybrids were determined by the projections from the positions of the hybrids onto the average tester ordinate (ATC ordinate) while the stability of the hybrids was measured by their projections from the average tester coordinate abscissa (ATC abscissa) or the horizontal line. The farther the hybrid from the ATC ordinate, the higher the yield while the shorter or closer the length of the projection of a hybrid to the ATC abscissa, the more stable the hybrid. Based on these criteria, the “mean vs. stability” of the GGE biplot view identified TZEQI 39 × TZEQI 44 (entry 25, a commercial check) as the highest yielding and TZEQI 210 × TZEQI 188 as the most stable QPM hybrid across research environments (Figure 4).

### 2.8. Step-Wise Multiple Regression and Sequential Path Analyses

Under the low-nitrogen environment, plant and ear aspects were identified by the stepwise multiple regression analysis as the first order traits, with significant contributions to grain yield and explaining 73% of the total variation in the grain yield of the QPM hybrids (Appendix A). Plant aspect had the highest direct negative contribution of −0.58 to grain yield. Five traits (stay green characteristic, days to 50% silking, ear height, ears per plant and plant height) contributed indirectly to grain yield through both plant and ear aspects and were identified as the second order traits. Among the second order traits, days to 50% silking made the highest positive (0.41) indirect contribution to grain yield through plant aspect. In contrast, ear height made the highest indirect negative contribution to grain yield through plant aspect. Days to 50% anthesis and ASI were identified as the third order traits. They contributed indirectly to grain yield through ears per plant, ear height, days to 50% silking and plant height. In the CDHS environment, ear aspect and ear height were identified as the traits with direct contributions to grain yield accounting for 49% of the total variation in grain yield (Appendix A). Ear aspect recorded the highest direct negative contribution (−0.54) to grain yield. Five traits were identified as the second order traits with plant aspect recording the highest indirect positive contribution to grain yield through the ear aspect. Among the third order traits, days to 50% anthesis had the highest indirect contribution to grain yield through plant aspect. Across stress environments, ear aspect and ear height were identified by the stepwise multiple regression analysis as the first order traits, with significant contributions to grain yield and explaining 55% of the total variation in the grain yield of the QPM hybrids (Figure 5). Ear aspect had the highest negative effect (−0.46) on grain yield. Seven traits (plant aspect, stay green characteristic, plant height, days to 50% silking, anthesis-silking interval, ears per plant, and root lodging) were identified as the second order traits. Among the seven traits, plant aspect had the highest indirect contribution of 0.58 and −0.61 to grain yield through ear aspect and ear height, respectively.

## 3. Discussion

The significant genotypic variation (*p* < 0.01) observed among the early QPM hybrids under low-nitrogen, CDHS, optimal and across test environments indicated that considerable genetic variation existed among the genotypes to facilitate accelerated gains from selection under contrasting environments. The observed significant differences for environments among the QPM hybrids revealed that each research environment was unique in terms of climatic and edaphic conditions which further suggested the need for extensive evaluations in contrasting environments. The significant genotype by environment interactions observed for grain yield and other measured traits under low-nitrogen, CDHS, optimal environments and across test environments implied that the hybrids responded distinctively to the environments. This emphasized that the environments were different and that the performance of the hybrids would not be consistent in different environments. Similar findings were reported by Ertiro et al. [12], Badu-Apraku et al. [26], Ifie et al. [30] under low-nitrogen and Meseka et al. [21], Nelimor et al. [22] under CDHS environments 

Very low heritability estimates were obtained for grain yield, anthesis and silking interval, and stay green characteristic of the QPM hybrids under low-nitrogen and tassel blasting under CDHS environments suggesting that selection based on phenotypic expression of these traits would be ineffective in achieving significant genetic gains. Contrary to the results of the present study, Meseka et al. [21] and Nelimor et al. [22,23] reported relatively moderate heritability estimates for grain yield under CDHS. Additionally, moderate to high repeatability estimates recorded for most measured traits indicated the reliability of the data for the measured traits in the present study [4].

Partitioning the QPM hybrids into genetic components, significant GCAm and GCAf mean squares were observed for grain yield and other measured traits under low-nitrogen environments implied that additive gene action was more important in the inheritance of grain yield and other measured traits under low-nitrogen. Similar findings were reported by Ifie et al. [30], Annor and Badu-Apraku [31], Obeng-Bio et al. [11], and Abu et al. [32] on the preponderance of the additive gene action over the non-additive gene action in the inheritance of grain yield under low-nitrogen environments. Contrarily, studies by Makumbi et al. [33], and Neolle et al. [34] reported non-additive gene action to be more important in the inheritance of grain yield under low-nitrogen environments.

Under CDHS environment conditions, the significant GCAf and SCA mean squares for grain yield and other measured traits, except for GCAm for grain yield, ASI, plant height, plant and ear aspects implied that both the additive and the non-additive gene actions were involved in the inheritance of grain yield and other measured traits under CDHS. Furthermore, the preponderance of SCA variances over GCA variances for grain yield indicated that non-additive gene action was more important than the additive and that the non-additive gene action was more important for the inheritance of grain yield traits under CDHS. This result is consistent with the findings of Chiuta and Mutengwa [24] who reported the preponderance of the non-additive gene action over the additive in the inheritance of grain yield under CDHS environments. Contrarily, Nasser et al. [29] reported the preponderance of the additive gene action over the non-additive in the inheritance of grain yield under CDHS environments. 

The significant GCAm, GCAf and the SCA mean squares for grain yield and other measured traits under optimal environments, except the GCAm for ASI and ear rot as well as the GCAf and the SCA for grain yield and ear rot, indicated that both the additive and the non-additive gene action were important in the inheritance of grain yield and other measured traits under optimal environments. The larger GCAm and GCAf mean squares for grain yield compared to the SCA mean squares, indicated that the additive gene action was more important in the inheritance of grain yield under optimal environments. This result is consistent with the findings of Makumbi et al. [33], Noelle et al. [34], Oyekale et al. [37], and Chiuta and Mutengwa [24] who reported that additive gene action controlled grain yield under optimal conditions.

The non-significant SCA mean squares recorded for the stay-green characteristic under low-nitrogen and CDHS environments indicated that the non-additive gene action was not important in the inheritance of the stay-green characteristic in this study. This result is similar to the findings of Badu-Apraku et al. [38] and Ifie et al. [30] who reported non-significant SCA mean squares for the stay-green characteristic under low-nitrogen conditions. However, the significant GCAm mean squares observed for the stay green characteristic under low-nitrogen and the significant GCAm and GCAf mean squares observed for the stay green characteristic under CDHS, indicated that there were variations in the expression of the QPM inbreds as parents in hybrid combinations for the stay green characteristic under CDHS conditions. Similar results were reported by Annor and Badu-Apraku [31] and Badu-Apraku et al. [26] who reported significant GCAm and GCAf effects for stay green characteristic under drought and low-nitrogen environments. The implications of these results are that recurrent selection methods should be the most effective for increasing the frequency of favourables alleles in a breeding population for the development of synthetic varieties. Contrarily, the preponderance of non-additive gene action implied that hybrid development should be employed under CDHS to exploit heterotic potentials of the QPM inbred lines. The differences in the results of the different authors could be attributed to differences in the germplasm studied, the statistical model employed, the severity of stress levels imposed and the experimental mating design used [4,26]. 

The non-significant GCAm × environment (E), GCAf × environment (E) and SCA × environment (E) interactions observed for measured traits except the GCAm × E for ASI and the GCAf × E for grain yield under low-nitrogen conditions, indicated that the performance of the parental lines were consistent across the low-nitrogen environments. This result is in agreement with the findings of Derera et al. [39] and Oyekale et al. [37] who reported non-significant GCAm × E, GCAf × E and SCA × E interactions for grain yield under drought and low-nitrogen environments. Contrarily, Annor and Badu-Apraku [31] and Badu-Apraku et al. [26] reported significant GCAm × E, GCAf × E and SCA × E interactions for grain yield and other measured traits of extra-early and early QPM inbreds under low-nitrogen stress. Similarly, Obeng-Bio et al. [11] reported significant GCAm × E, GCAf × E and SCA × E interactions for grain yield and other traits in early pro-vitamin A (PVA) QPM inbred lines under low-nitrogen. The significant GCAf × E interaction for grain yield suggested that the GCA effects associated with the female parents were not the same across the low-nitrogen environments. 

Under low-nitrogen environments, comparable contributions of GCAm and GCAf sum of squares were observed for grain yield and other measured traits except ASI, indicating that both maternal and paternal effects were equally important in the inheritance of the measured traits of the hybrids. Similar studies by Ifie et al. [30], Annor and Badu-Apraku [31], Obeng-Bio et al. [11] reported no significant differences in the contribution of both GCAm and GCAf (paternal and maternal effects) for grain yield and other measured traits under low-nitrogen. However, GCAm for ASI was larger than GCAf, suggesting the importance of paternal effects in the inheritance of anthesis and silking interval. The larger GCAf sum of squares relative to GCAm sum of squares observed under CDHS environments for grain yield and plant aspect indicated the greater importance of cytoplasmic effect on the inheritance of grain yield and plant aspect. Also, ears per plant had significantly larger GCAf sum of squares than the GCAm sum of squares indicating that maternal effects conditioned prolificacy and that the parental lines with significant and positive GCAf effects for ears per plant should be used as the female parents in hybrid production to ensure prolificacy under CDHS.

According to Girma et al. [40], the information on the combining of ability of parental lines for a trait is useful in determining the contribution of the parental lines to their progenies in hybrid combinations. In other words, parental lines with significant GCA effect for a trait under a stress condition has high probability of transferring the favourable alleles for the trait to progenies in hybrid combinations and such parental lines could be useful in the development of outstanding hybrids [4]. The observed significant and positive GCAm and GCAf effects for grain yield recorded by TZEQI 6 under low-nitrogen environments indicated that the parental line would transfer favourable alleles to the progeny when used either as a female or a male parent. Similarly, the significant and positive GCAm effects for grain yield recorded by TZEQI 106, TZEQI 113, TZEQI 158, TZEQI 188 and TZEQI 6 under low-nitrogen environments suggested that these inbred lines when used as male parents, would transfer favourable alleles for grain yield to their progenies under low-nitrogen environments. A similar inference could be made for the inbred TZEQI 210 which displayed significant and positive GCAf effects for grain yield. Additionally, the observed significant and positive GCAm effects for grain yield by TZEQI 113 and TZEQI 216 implied that both parental lines would transfer favourable alleles to their progenies when used as male parents under CDHS environment. In addition, the significant and negative GCAf effects observed for stay green characteristic was an indication that TZEQI 6, TZEQI 113, TZEQI 132, TZEQI 158 and TZEQI 219 would contribute favourable alleles to their progenies for delayed senescence or prolonged stay green characteristic, increased photosynthesis and assimilate production under low-nitrogen conditions.

Using the HGCAMT grouping method, the 24 QPM inbred lines were classified into four heterotic groups. Interestingly, crosses between inbred lines from different heterotic groups displayed higher heterosis across stress environments indicating the effectiveness of the grouping method. The heterotic groups identified by the HGCAMT method across environments would increase the chances of developing outstanding early maturing QPM hybrids and synthetics with CDHS and low-nitrogen tolerance for commercialization in SSA. Additionally, heterotic populations could be developed by recombining QPM inbred lines from the same heterotic group and improving the population through recurrent selection methods [41].

Based on the criteria proposed by Pswarayi and Vivek [36] for identifying inbred and single-cross testers, inbred line TZEQI 6 with significant and positive GCAm and GCAf effects for grain yield was identified as a tester under low-nitrogen environment conditions, suggesting that TZEQI 6 was a good combiner and could be used either as a male or a female parent in the development of high yielding hybrids and for grouping of other inbred lines under low-nitrogen environments. This result confirmed the earlier report by Badu-Apraku and Fakorede [4] who identified TZEQI 6 as a tester. It was therefore, not surprising that TZEQI 6 was involved in the following outstanding hybrid combinations: TZEQI 6 × TZEQI 228, TZEQI 113 × TZEQI 6, TZEQI 6 × TZEQI 210 and TZEQI 6 × TZEQI 219. Hybrid TZEQI 113 × TZEQI 6 was identified as a single-cross tester and could be used for the development of three-way and double-cross hybrids [26].

An important objective of this study was to identify the most outstanding hybrids under each and across test environment conditions. The QPM hybrids TZEQI 6 × TZEQI 219 and TZEQI 6 × TZEQI 228 were identified as the most outstanding across CDHS and low-nitrogen environments. These hybrids could be useful in the development of early maturing multiple-stress tolerant three-way QPM hybrids for commercialization in SSA. 

The significant genotype by environment interaction observed for grain yield under CDHS, low-nitrogen and across test environments in the present study justified the use of the GGE biplot to identify high yielding and stable QPM hybrids for commercialization in SSA. From the “which-won-where” view of the GGE biplot, hybrids TZEQI 241 × TZEQI 216, TZEQI 6 × TZEQI 210, TZEQI 39 × TZEQI 44 (check) and TZEQI 6 × TZEQI 55 (check) were identified as the highest yielding in low-nitrogen, optimal and CDHS environments, respectively. Additionally, the mean yield vs. stability of the GGE biplot view identified TZEQI 39 × TZEQI 44 (check) as the highest yielding and TZEQI 210 × TZEQI 188 as the most stable QPM hybrids across test environments. These results suggested that these hybrids would display superior performance in nitrogen depleted, drought prone environments without compromising grain yield.

Because of the quantitative nature of grain yield and the low heritability recorded for grain yield under the stress environment conditions used in this study, selection using grain yield alone without other agronomic traits would be ineffective [42]. Therefore, there was the need to examine the interactions among the secondary traits and their relative contribution to grain yield for effective selection. Under low-nitrogen conditions, plant and ear aspects were identified as the most important secondary traits contributing to the observed variations in grain yield, thus suggesting their reliability as secondary traits for selection under low-nitrogen environments. In an earlier study, Badu-Apraku et al. [43] using sequential path analysis, identified plant and ear aspects as the first order traits with direct contribution to grain yield under low-nitrogen environments. Additionally, the stay green characteristic contributed indirectly to grain yield through both plant and ear aspects, indicating its reliability as a secondary trait thus justifying its inclusion in the selection base index under low-nitrogen environments. Under CDHS environments, the direct contributions of both ear aspect and ear height to grain yield indicated that they were the major determinants of grain yield. Plant aspect contributed indirectly to grain yield through the two first order traits (ear aspect and ear height), indicating its importance in the selection of genotypes under CDHS environments. Similarly, both ear aspect and ear height had direct contributions to grain yield across test environments, implying that they were the major contributors to grain yield. These results justified the inclusion of plant and ear aspects in the selection index under both stress conditions. Additionally, this suggested the need to revisit the issue of the inclusion of ear height as one of the secondary traits in the selection index under stress environments. However, increasing ear height is not desirable because it is associated with stalk lodging. Therefore, in the IITA-MIP plant and ear heights are usually pegged and not allowed to increase.

## 4. Materials and Method

### 4.1. Genetic Materials and Testcrosses

The genetic materials used in this study comprised twenty-four early white QPM inbred lines extracted from F_1_ maize hybrids of ten bi-parental crosses involving crosses among extra-early white QPM inbred testers and early maturing white QPM inbred testers. The QPM inbred line testers were identified to have positive and significant general combining ability from previous studies [4,26] (the F_1_ hybrids were taken through a cycle of backcrossing to the extra-early inbred testers to recover the earliness. The BC_1_F_1_ with desirable agronomic characteristics were selected using pedigree selection from each backcrossed population and advanced through repeated inbreeding to the S_7_ generation while selecting for kernels with the appropriate endosperm modification ranging from 25–50% (Table 1). The 24 inbred lines were selected based on their reactions to low-nitrogen and CDHS in the preliminary evaluations (Table 6). The inbred lines were inter-mated to generate 96 F1 hybrids using the North Carolina II mating Design (NCDII) proposed by Comstock and Robinson [44]. This was achieved by dividing the QPM inbred lines into six sets each comprising four early QPM inbred lines. Each inbred line was used as a male in one set and as a female in another set, to produce a total of six sets each containing four inbred lines, resulting in a total of 96 design II single-cross hybrids (Appendix A).

### 4.2. Experimental Sites and Field Evaluation

The 96 early white QPM hybrids generated from the crosses of the 24 QPM inbred lines and four released IITA commercial hybrid checks (Appendix A) were evaluated using a 10 × 10 lattice design with two replications at the IITA experimental station, Mokwa, Nigeria (9°18′ N, 5°4′ E, 457 m altitude, 1100 mm annual rainfall) during the 2019 and 2020 rainy season (between June and October). The low-nitrogen experiment was carried out at Mokwa where the soil had been depleted of nitrogen by the continuous growing of maize without the application of fertilizer and the removal of the biomass after each cropping season. Therefore, the low-nitrogen blocks were depleted of nitrogen to zero level. During each season, three seeds were planted per hill and seedlings were thinned to two plants per hill 2 weeks after planting (WAP) to obtain the target population density of 66,666 plants ha^−1^. The seeds were planted in single-row plots of 3 m long with spacing of 0.75 m and 0.40 m between and within rows, respectively. The nitrogen fertilizer in the form of urea (30 kg N ha^−1^) and 15 kg N ha^−1^ were applied at 2 WAP with additional 15kg N ha^−1^ applied at 4 WAP. The low-nitrogen plots received 60 kg ha^−1^ each of single superphosphate (P2O5) and muriate of potash (K_2_O) at 2 WAP. The low-nitrogen plots were kept weed-free with the application of atrazine and gramozone as pre- and post-emergence herbicides at 5 L/ha, respectively, and subsequently supplemented with hand weeding to keep the plots weed-free.

The 96 early white QPM hybrids plus four hybrid checks were also evaluated for their agronomic performance under CDHS conditions at the IITA experimental station in Kadawa (11°45′ N, 8°45′ E, 468.5 m ASL, 884 mm annual rainfall) during the 2020 and 2021 dry seasons, where extreme drought stress at high temperatures between 33 and 45 °C occurred between February and June every year [20] (Appendix A). The CDHS trials were irrigated twice every week for the first 28 days after planting using a furrow irrigation system with the plants relying on the stored soil moisture. The plants were subjected to CDHS for 3 weeks during the month of April when the day temperature varied from 35 to 40 °C with the night temperature ranging from 22 to 28 °C [4,20]. A 10 × 10 alpha-lattice design with two replicates were used for the evaluation of the 100 Design II single-cross hybrids. The experimental units were one-row plots; each 3 m long with row spacing of 0.75 m and the distance between two adjacent plants within the rows were 0.40 m in all trials. The trials were kept weed free through the application of atrazine and gramozone as pre- and post-emergence herbicides supplemented with manual weeding.

The QPM hybrids were also evaluated using 10 × 10 lattice design with two replications respectively under optimal conditions at Mokwa and Kadawa during the 2019/2020 (between June and October) rainy seasons. Fertilizer was applied at the rate of 30 kg N ha^−1^, 60 kg ha^−1^ each of P2O5 and K_2_O at 2 WAP and additional 60 kg ha^−1^ N was top-dressed using urea at 4 WAP. The trials were kept weed free as described in the other trials.

### 4.3. Data Collection

Data were collected under each research conditions for number of days to 50% silking (DS) and anthesis (DA). Anthesis-silking interval (ASI) was calculated as the difference between DS and DA. Plant height (PLHT) was measured as the distance from the base of the plant to the height of the first tassel branch. Ear height (EHT) was measured in centimetres as the distance from the base of the plant to the node bearing the upper ear. Root lodging (RL) was measured as the percentage of plants leaning more than 30°(degree) from the vertical whereas stalk lodging (SL) was recorded as the percentage of plants broken at or below the highest ear node. Plant aspect (PASP) was recorded on a scale of 1 to 9 based on plants appeal to sight, where 1 = excellent phenotypic appeal and 9 = extremely poor phenotypic appeal. Ear aspect (EASP) assessed the freedom from disease and insect damage; ear size and uniformity of ears was scored on a scale of 1 to 9, where 1 = large, uniform, clean, and well-filled ears and 9 = ears with totally unattractive features. Number of ears per plant (EPP) was calculated by dividing the total number of ears harvested per plot (EHARV) by the number of plants in a plot at harvest (PHARV). Ear rot (EROT) was computed as the percentage of ears harvested that had ear rot. Stay green (STGR) characteristic was scored at 70 DAP on a 1 to 9 scale as percentage of dead leaf area under low-nitrogen and CDHS conditions, where, 1= all leaves are green and 9 = all leaves are dead. Data on tassel blast (TABLAST) and leaf firing were recorded during flowering on the CDHS trial. TABLAST was rated during flowering on a scale of 1 to 9, where 1 = all plants had normal pollen production and 9 = all plants had white, dry tassels without pollen production and showed severe tassel blast. 

### 4.4. Data Analysis

Data obtained for grain yield and other measured agronomic traits under low-nitrogen and CDHS environments were subjected to analysis of variance (ANOVA) with PROC GLM statement in SAS version 9.4 [45] to obtain mean squares for each trait. Each year-location combination constituted a test environment. In the combined ANOVA, test environments, replications, genotype × environment interactions, and all other sources of variation were considered as random effects while genotypes were considered as fixed effects. The NCD II analysis of variance for each environment was conducted on the genotypes excluding the checks using PROC GLM statement in SAS [44]. The hybrids component of variation was partitioned into sources of variation and main effects as male(sets) (GCAm), female(sets) (GCAf), and female × male(sets) (SCA) interaction as described by Hallauer and Miranda [46]. The F tests for male(sets), female(sets), and female × male(sets) mean squares were estimated using the mean squares of the respective interactions with environment. The mean squares attributable to male × female (sets) × environment were tested using the pooled error mean squares. The broad-sense heritability (H^2^) estimates for each measured trait was calculated for each measured traits under each environment, and across the environments were estimated as follows:$$H^{2}=\frac{\sigma_{g}^{2}}{\sigma_{g}^{2}+\frac{\sigma_{e}^{2}}{r}}$$
where
$\sigma_{g}^{2}$ is the genotypic variance, 
$\sigma_{e}^{2}$
is experimental error variance; and r is the number of replicates within each test environment [47]. Repeatability estimates were assessed as the consistency of the expression of the measured traits across the different environments and the data from a location with repeatabilty estimate of below 30% was discarded as not reliable [4,48]. Thus, repeatability estimates were calculated on a genotype-mean basis as follows:$$R=\frac{\sigma_{g}^{2}}{\sigma_{g}^{2}+\frac{\sigma_{ge}^{2}}{e}+\frac{\sigma^{2}}{re}}$$
where $\sigma_{g}^{2}$ is the genotypic variance, $\sigma_{ge}^{2}$ is the variance attributable to genotype by environment interaction, $\sigma^{2}$ is the error variance, *r* is the number of replication and *e* is the number of environments.

### 4.5. The Proportionate Contribution of Combining Ability

The proportionate contribution for each measured trait was calculated as the percentage of the sum of squares for the crosses attributed to general combining ability (GCA) and specific combining ability (SCA). The general combining ability (GCA) effects for male and female within sets (GCAm and GCAf) for each measured trait were computed from the adjusted means using line × tester approach [49]. As shown below: GCAf = X_f_ − µ, 
GCAm = X_m_ − µ 
where, GCAm and GCAf were the general combining ability effects of male and female parents respectively, X_f_ and X_m_ were the mean performance of the male and female parents respectively and µ is the overall mean of crosses in the trial. 

The standard errors for GCA effects were calculated as described by Cox and Frey [50]:SE(GCAf) = [MS_fe_ (f^−1^)/*f* × *e* × *r*]^1/2^
SE(GCAm) = [MS_me_ (m^−1^)/*m* × *e* × *r*]^1/2^
where, MS_fe,_ and MS_me_ were the mean squares of the interactions between male and female with the environment; MS_fme_ = was the mean squares of the female × male by environment interaction. *m*, *f*, *e*, and *r* are the numbers of male, females, environments, and replicates, respectively.

### 4.6. Heterotic Grouping of the Inbred Lines

Classifying newly developed early maturing QPM inbred lines into appropriate heterotic groups is important for the exploitation of maximum heterosis through crossing of inbreds from opposing heterotic groups [41]. The 24 inbred lines were classified into heterotic groups using the heterotic grouping method based on GCA effects of multiple traits (HGCAMT) method proposed by Badu-Apraku et al. [26]. The GCA effects of traits with significant mean squares for genotype under low-nitrogen, CDHS, optimal and across stress conditions were standardized to minimize the effects of different traits scale. The standardized GCA effects values were subjected to Ward’s minimum variance cluster analysis using SAS version 9.4 [45] as described by Badu-Apraku et al. [26].

### 4.7. Identification of Inbred and Single-Cross Testers

An inbred line was identified as a tester based on the following criteria proposed by Pswarayi and Vivek, [36]: it should (i) belong to a heterotic group; (ii) have a significant and positive GCA effects across the test environments and (iii) have a high per-se yield performance. To identify a single-cross tester, the parental lines of the hybrid must: (i) record significant and positive GCA effects for grain yield, (ii) the parental lines of the hybrid should belong to the same heterotic group, and (iii) the hybrid should have relatively good yielding ability under stress conditions. To identify promising hybrids with high grain yield and tolerance to low soil nitrogen and CDHS among the hybrids, a multiple trait base index (MI) proposed by Badu-Apraku et al. [26,51] was adopted. The MI integrated grain yield, number of ears per plant, plant aspect, ear aspect, anthesis silking interval as well as the stay green characteristic as follows:Multiple trait base index (MI) = [(2 × GYLD) + EPP − ASI − PASP − EASP − STGR]
where: GYLD = grain yield, EASP = ear aspect, ASI = anthesis-silking interval, EPP = ears per plant, PASP = plant aspect, and STGR = stay green characteristic for the low-nitrogen and the CDHS environments. The parameters used in MI were standardized to reduce the effects of different scales of the parameters. A positive value of the MI was an indicator of CDHS and low-nitrogen tolerance, while a negative value indicated hybrid susceptibility.

### 4.8. Stability of Hybrids across Test Environments

A set of 22 QPM hybrids (top 12 and worst 10) including the four QPM hybrid checks were selected under CDHS and low-nitrogen conditions using the multiple trait selection base index values for the genotype main effect plus genotype by environment interaction (GGE) biplot analysis to separate the genotype by environment interactions into the component parts [52,53]. The GGE biplot was used to obtain information on the high yielding and most stable hybrids across test environments using the genotype by environment analysis with R for Windows (GEA-R) software [54]. The “mean vs. stability” view of the GGE biplot was employed to identify hybrids with high grain yield and stability across CDHS, low-nitrogen and optimal environments. The GGE biplot model equation used was as follows:$$Y_{ij}-Y_{j}=\lambda_{1}\varepsilon_{i1}\eta_{j1}+\lambda_{2}\varepsilon_{i2}\eta_{j2}+\varepsilon_{ij}$$
where, *Y_ij_* is the average yield of genotype *i* in environment *j*, *Y_j_* is the average yield across genotypes in environment *j*, $\lambda_{1}$ and $\lambda_{2}$ are the singular values for PC1 and PC2 respectively, $\varepsilon_{i1}$ and $\varepsilon_{i2}$ are the PC1 and PC2 scores for genotype *i*, $\eta_{j1}$ and $\eta_{j2}$ are the PC1 and PC2 scores for genotype *j*, $\varepsilon_{ij}$ is the error associated with the genotype *i* in environment *j*.

### 4.9. Inter-Trait Relationships under CDHS and Low-Nitrogen Environments

The step-wise multiple regression and sequential path diagrams were employed to determine the causal relationships between the measured traits under CDHS and low-nitrogen environments using the procedure described by Mohammadi et al. [55]. The step-wise multiple regression analysis was done using the Statistical Package for the Social Sciences, SPSS v. 17.0 [56] to determine the first, second, third, and fourth order predictor traits on the basis of their contributions to grain yield variation. The secondary traits were regressed on grain yield to identify first order traits that contributed significantly to grain yield at *P* ≤ 0.05. The remaining secondary traits were regressed on the first order traits to identify those with significant contributions to grain yield and they were categorized as second order traits. The procedure was repeated in order to categorize the remaining traits into subsequent orders. The standardized *b* values generated by the step-wise regression analysis were the path coefficients [42,55]. The significance of a path coefficient was determined in the stepwise regression analysis using the t-test with a probability level of 5% and only traits with a significant path coefficients were retained. In addition, spearman correlation analysis implemented in SAS v.9.4 [45] was done to determine the relationships among traits within the same order.

## 5. Conclusions

There was preponderance of additive gene action over non-additive one in the inheritance of grain yield under low-nitrogen and optimal environment conditions. The implication is that yield of maize hybrids under low-nitrogen and optimal environment conditions could be enhanced through recurrent selection and that inbred lines tolerant to low-nitrogen with high combining ability effects could be extracted from improved cycles of selection for hybrid development. Under CDHS environments, the non-additive gene action was more important than the additive gene action in the inheritance of grain yield implying that hybrid development should be employed under CDHS to exploit heterosis. Additionally, maternal effects influenced the inheritance of grain yield under CDHS environments. Plant aspect, ear aspect and ear height were identified as the primary traits contributing to the observed variations in grain yield under CDHS and low-nitrogen conditions. TZEQI 6 was identified as an inbred tester while hybrid TZEQI 113 × TZEQI 6 was identified as a single-cross tester. This implied that TZEQI 6 could be used either as a male or a female parent in the production of high yielding hybrids while hybrid TZEQI 113 × TZEQI 6 could be adopted as a single-cross tester for a three-way or double-cross hybrid production under low-nitrogen environments. Hybrids TZEQI 6 × TZEQI 228 and TZEQI 210 × TZEQI 188 should be tested extensively in on-farm trials and commercialized in SSA as combined low-nitrogen and CDHS tolerant hybrids.

## Figures and Tables

**Figure 1 plants-10-02596-f001:**
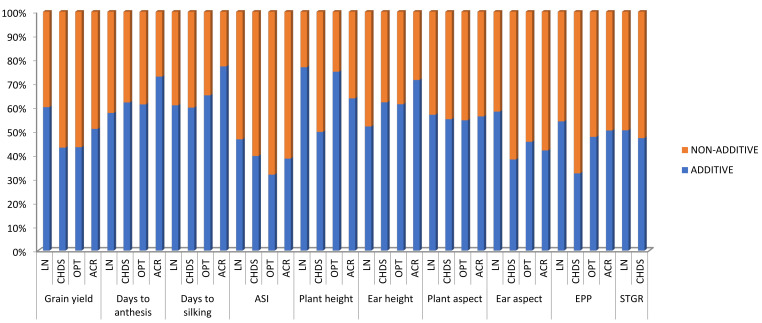
Proportion of additive and non-additive genetic variances for grain yield and other agronomic traits under low-nitrogen, CDHS, optimal conditions and across test environments.

**Figure 2 plants-10-02596-f002:**
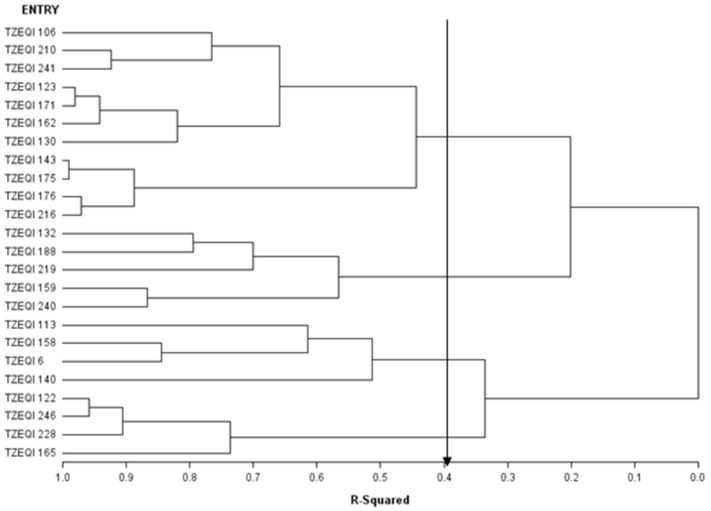
Heterotic grouping of the 24 early white QPM inbred lines.

**Figure 3 plants-10-02596-f003:**
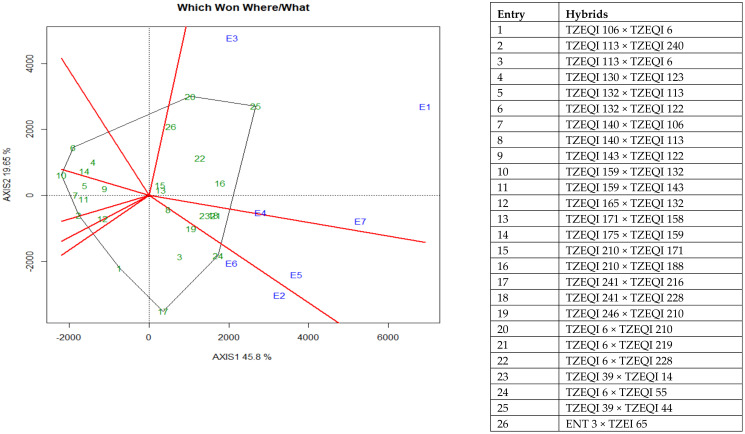
A “which won where” genotype plus genotype by environment interaction biplot of grain yield of selected 22 early QPM hybrids plus checks based on CDHS and the low-nitrogen base index evaluated across CDHS, low N and optimal environments 2019 and 2021. E1 = Mokwa under low-nitrogen environment, 2019, E2 = Mokwa under low-nitrogen environment, 2020, E3= Mokwa under optimal environment, 2019, E4 = Mokwa optimal environment, 2020, E5 = Kadawa optimal environment, 2020, E6 = Kadawa under CDHS environment, 2020, E7 = Kadawa under CDHS environment, 2021.

**Figure 4 plants-10-02596-f004:**
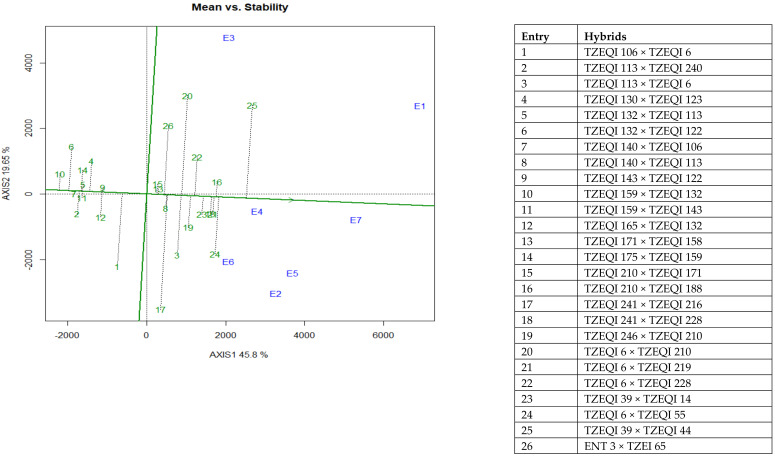
An entry/tester genotype main effect plus genotype by environment interaction biplot of grain yield of selected 22 early QPM hybrids plus checks based on CDHS and low-nitrogen base index evaluated across CDHS, low N and optimal environments from 2019 and 2021. E1 = Mokwa under low-nitrogen environment, 2019, E2 = Mokwa under low-nitrogen environment, 2020, E3= Mokwa under optimal envionment, 2019, E4 = Mokwa optimal environment, 2020, E5 = Kadawa optimal environment, 2020, E6 = Kadawa under CDHS environment, 2020, E7 = Kadawa under CDHS environment, 2021.

**Figure 5 plants-10-02596-f005:**
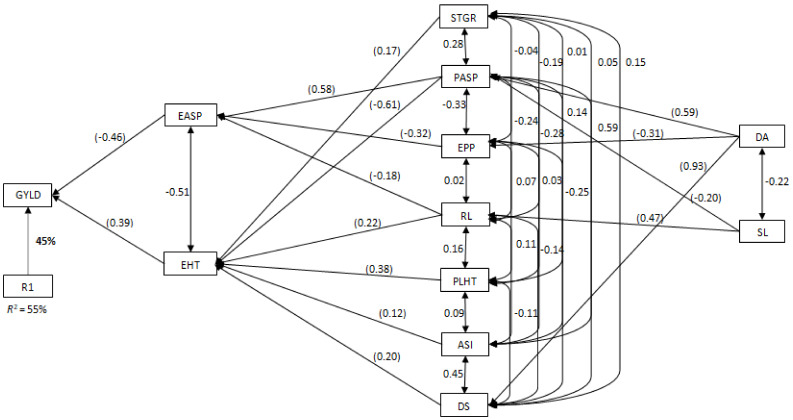
Path analysis diagram showing inter-trait relationships of traits of early QPM hybrids evaluated across test environments in Nigeria. Bold value is the residual effect; direct path coefficients values in parenthesis while correlation coefficients are the other values. R^2^ = coefficient of determination; R1= residual effects; GYLD = Grain yield; ASI = anthesis-silking interval; PASP = plant aspect; PLHT = plant height; EASP = ear aspect; EHT = ear height; EPP = ears per plot; STGR = stay green characteristic; DA= days to 50% anthesis; RL = root lodging; DS = days to 50% silking; SL = stalk lodging; TABLAST= tassel blast.

**Table 1 plants-10-02596-t001:** Mean squares of grain yield and other agronomic traits of 96 early QPM hybrids evaluated under low-nitrogen conditions at Mokwa 2019 and 2020 growing seasons and under CDHS at Kadawa during the 2020 and 2021 dry seasons.

SV	DF	YIELD	DA	DS	ASI	PLHT	EHT	PASP	EASP	STGR	TABLAST
**CDHS environment**											
ENV	1	595549601.1 **	4.190 **	3.45 **	0.02 **	1021.33ns	4365.28 **	9.20 **	69.69 **	32.77 **	2.45 **
SET	5	90806981.5 **	42.08 **	28.27 **	4.18 **	3402.94ns	2542.63 **	10.94 **	14.09 **	1.30ns	0.11ns
ENV*SET	5	95851846.7 **	12.14 **	4.69ns	4.27 **	3008.44ns	61.51ns	7.72 **	1.61ns	1.13ns	0.10ns
Rep(ENV*SET)	10	3262635.4ns	4.83ns	5.55ns	0.66ns	1965.49ns	33.84ns	1.36ns	0.86ns	0.70ns	0.08ns
Block(ENV*Rep)	36	6681421.9 **	16.47 **	22.09 **	1.66 *	3090.16ns	165.32 **	3.89 **	2.95 **	3.32 **	0.12 **
HYBRID	99	1419012 **	8.53 **	9.98 **	1.26ns	4279.06ns	323.48 **	1.96 **	2.86 **	1.07 *	0.12 **
GCA_MALE_(SET)	18	1666352.2ns	9.92 **	12.88 **	1.14ns	5610.35ns	311.05 **	1.20ns	2.01ns	1.32 *	0.14 **
GCA_FEMALE_(SET)	18	6944283.3 **	10.65 **	13.13 **	1.24ns	5084.04ns	269.36 **	2.42 **	1.95ns	1.37 *	0.08ns
SCA(SET)	54	3790544.1 *	4.20 *	5.81ns	1.21ns	3610.24ns	118.18 **	0.99ns	2.14ns	1.01ns	0.14 **
ENV*HYBRID	99	781347.6 **	4.74 **	6.38 **	1.47 *	3898.23ns	64.11ns	1.86 **	2.58 **	1.16 *	0.11 **
ENV*GCA_MALE_(SET)	18	2723378.3ns	2.67ns	2.97ns	1.42ns	4486.43ns	79.26ns	1.43ns	4.36 **	1.13ns	0.14 **
ENV*GCA_FEMALE_(SET)	18	6455338.8 **	3.48ns	6.88 *	1.57ns	4644.67ns	48.45ns	2.11 *	3.36 **	1.38 *	0.07ns
ENV*SCA(SET)	54	4069373.8 **	5.54 **	8.02 **	1.33ns	3588.54ns	63.04ns	1.35ns	1.47ns	1.18 *	0.10 **
Error	143	2517929	2.95	4.19	1.06	3995.32	57.15	1 1.14	1.56	0.78	0.05
H^2^	-	0.37	0.54	0.50	0.27	0.39	0.74	0.41	0.51	0.26	0.31
Repeatability	-	0.53	0.47	0.39	0.44	0.64	0.79	0.43	0.60	0.41	0.62
**Low-nitrogen environment**											
ENV	1	2092131.3 *	2218.83 **	1492.62 **	256.67 **	16102.05 **	2585.48 **	56.80 **	39.09 **	27.589 **	-
SET	5	4087462.8 **	5.39ns	1.14ns	1.49ns	3901.85 **	1005.06 **	0.7 **	1.71 **	1.70 *	-
ENV*SET	5	2249671.8 **	8.88ns	9.11ns	0.47ns	214.05ns	176.55ns	0.15ns	0.39ns	1.02ns	-
Rep(ENV*SET)	10	458860ns	4.48ns	3.93ns	3.12 **	133.83ns	199.94ns	0.18ns	0.44ns	1.11ns	-
Block(ENV*Rep)	36	5795924.1 **	12.51 **	27.29 **	2.27 *	1222.36 **	510.66 **	2.80 **	3.63 **	3.07 **	-
HYBRID	99	977907.6 **	5.54 **	7.12 **	0.51ns	481.69 **	178.59 **	2.26ns	1.92 **	1.92 **	-
GCA_MALE_(SET)	18	1129415.8 **	7.73 **	12.78 **	1.28 **	545.15 **	156.09ns	0.64 *	0.46ns	0.8ns	-
GCA_FEMALE_(SET)	18	952771.1 *	6.01 *	10.60 **	0.72ns	452.67 **	171.41ns	0.31 *	0.53 **	0.37 *	-
SCA(SET)	54	463614.4ns	3.93ns	5.05ns	0.88ns	100.34ns	101.36ns	0.32ns	0.34ns	0.77ns	-
ENV*HYBRID	99	977907.6 **	5.54 **	7.12 **	0.51ns	481.69 **	178.59 **	2.26 **	1.92 **	1.92 **	-
ENV*GCA_MALE_(SET)	18	684249ns	4.53ns	5.06ns	1.70 **	139.48ns	101.64ns	0.55ns	0.56ns	0.69ns	-
ENV*GCA_FEMALE_(SET)	18	1254953.7 **	5.48ns	4.16ns	1.18ns	114.92ns	126.85ns	0.29 *	0.63 **	0.71ns	-
ENV*SCA(SET)	54	564756.6ns	3.66ns	5.64ns	0.93ns	139.4ns	87.42ns	0.31ns	0.43ns	0.87ns	-
Error	143	560355.6	3.49	4.76	0.76	151.37	109.3	0.39	0.54	0.73	-
H^2^	-	0.37	0.33	0.34	0.23	0.52	0.44	0.21	0.12	0.20	-
Repeatability	-	0.71	0.66	0.68	0.58	0.83	0.69	0.63	0.56	0.58	-

*, ** Significant at 0.05 and 0.01 probability levels, respectively; ns = not significant; ENV = environment; Rep = replication; YIELD = Grain yield; DA = days to 50% anthesis; DS = days to 50% silking; ASI = antheisi-silking interval; PLHT = plant height; EHT = ear height; PASP = plant aspect; EASP = ear aspect; STGR = stay green characteristic; TABLAST = tassel blast.

**Table 2 plants-10-02596-t002:** Mean squares of grain yield and other agronomic traits of 96 early QPM hybrids evaluated under optimal conditions at Mokwa and Kadawa during 2019 and 2020 growing seasons and across test environments.

SV	DF	YIELD	DA	DS	ASI	PLHT	EHT	PASP	EASP	EROT
**Optimal environments**										
ENV	2	654194407 **	11058.25 **	12565.68 **	196.41 **	431852.40 **	110406.77 **	82.65 **	45.28 **	1053.99 **
SET	5	19427073 **	9.54 **	6.97ns	2.62 **	9601.23 **	2308.62 **	10.21 **	11.46 **	1.36ns
ENV*SET	10	8473814ns	19.75 **	27.58 **	1.02ns	201.49 **	177.39 **	0.61ns	1.04ns	1.21ns
Rep(ENV*SET)	15	2900486ns	2.13ns	5.47ns	1.22 **	116.18ns	96.09ns	0.48ns	0.78ns	1.80ns
Block(ENV*Rep)	54	6629211ns	6.56 **	6.17 **	0.73ns	480.07 **	324.82 **	0.55 **	1.13 **	4.62 **
HYBRID	99	5711483ns	12.52 **	10.35 **	1.10 **	940.42 **	312.43 **	1.47 **	1.69 **	2.32ns
GCA_MALE_(SET)	18	8462986 **	10.74 **	9.98 **	0.45ns	632.78 **	294.88 **	0.90 **	1.09 **	1.84ns
GCA_FEMALE_(SET)	18	2666167ns	22.88 **	18.30 **	1.00ns	823.93 **	322.12 **	1.43 **	1.08 **	2.94ns
SCA(SET)	54	4859033ns	7.10 **	5.07 **	1.04 **	162.70 **	130.02 **	0.65 **	0.87 **	2.26ns
ENV*HYBRID	198	5182385ns	6.97 **	7.62 **	1.00 **	190.53 **	146.94 **	0.57 **	0.71 *	2.31ns
ENV*GCA_MALE_(SET)	36	6852042ns	6.31 **	7.42 **	1.18 **	304.61 **	180.37 **	0.39ns	0.79ns	2.38ns
ENV*GCA_FEMALE_(SET)	36	5273256ns	6.77 **	9.46 **	0.96 **	202.93 **	211.50 **	0.62 **	0.58ns	2.95 **
ENV*SCA(SET)	108	4299183ns	5.20 **	5.33 **	0.99 **	148.51 **	116.89 **	0.57 **	0.71ns	2.19ns
Error	215	4941851	3.57	3.54	0.62	100.2	88.35	0.4	0.57	1.97
H^2^	-	0.41	0.48	0.53	0.38	0.78	0.6	0.58	0.51	-
Repeatability	-	0.65	0.54	0.59	0.43	0.81	0.56	0.62	0.61	-
**Across environments**										
ENV	6	499379987 **	13849.13 **	15593.72 **	123.97 **	154513.34 **	45040.16 **	46.41 **	54.38 **	557.78 **
SET	5	25520360 **	34.39 **	12.66 **	5.60 **	14753.19 **	5416.08 **	17.53 **	22.77 **	2.59ns
ENV*SET	30	3762534 *	13.86 **	15.39 **	1.52 **	942.61ns	171.52 **	2.28 **	1.45 *	1.82ns
Rep(ENV*SET)	35	1608595ns	3.58ns	5.06ns	1.59 **	649.6ns	107.98ns	0.65ns	0.71ns	1.42ns
Block(ENV*Rep)	126	4851705 **	11.09 **	16.75 **	1.41 **	1437.89	332.34 **	2.15 **	2.37 **	5.77 **
HYBRID	99	4126473 **	15.01 **	15.52 **	1.40 **	3033.39 **	605.93 **	2.41 **	2.74 **	2.94 **
GCA_MALE_(SET)	18	4424226 **	19.75 **	24.77 **	1.52 *	3653.58 **	532.65 **	1.21 **	1.16ns	4.11 **
GCA_FEMALE_(SET)	18	2103577 *	24.74 **	25.50 **	0.69ns	3564.40 **	586.29 **	2.50 **	1.57 *	3.90 **
SCA(SET)	54	2094763ns	5.53 **	4.97ns	1.18 *	1368.91ns	149.13 **	0.97 **	1.26 *	1.72ns
ENV*HYBRID	594	2629285 **	5.86 **	6.56 **	1.14 **	1173.7ns	111.95 **	0.81 **	1.20 **	2.31 **
ENV*GCA_MALE_(SET)	108	3505822 *	4.67 *	5.64 *	1.13 **	1443.02ns	127.85 **	0.71ns	1.46 **	1.96
ENV*GCA_FEMALE_(SET)	108	2453356 **	6.25 **	7.74 **	1.13 **	1357.17ns	130.50 **	0.86 *	1.21 **	2.84 **
ENV*SCA(SET)	323	2295330ns	4.93 **	5.98 **	1.05 **	1085.28ns	97.5ns	0.64ns	0.94ns	2.30 *
Error	501	2474083	3.37	4.08	0.8	1226.59	85.42	0.61	0.85	1.9

*, ** Significant at 0.05 and 0.01 probability levels, respectively; ns = not significant; ENV = environment; Rep = replication; YIELD = Grain yield; DA = days to 50% anthesis; DS = days to 50% silking; ASI = antheisi-silking interval; PLHT = plant height; EHT = ear height; PASP = plant aspect; EASP = ear aspect; EROT = ears rot.

**Table 3 plants-10-02596-t003:** Proportion of the sums of squares attributable to general combining ability (GCA) and specific combining ability (SCA) for grain yield and other agronomic traits of early QPM inbred lines under low-nitrogen, CDHS, optimal and across test environments.

TRAITS	Low-Nitrogen			CDHS			Optimal Conditions			Across Test Environments		
	GCA		SCA	GCA		SCA	GCA		SCA	GCA		SCA
	Male	Female		Male	Female		Male	Female		Male	Female	
Grain yield (kg ha^−1^)	32.5	27.5	39.9	8.3	34.8	56.9	32.9	10.4	56.7	34.5	16.4	49.0
Days to 50% anthesis	32.6	25.0	42.4	29.9	32.1	38.0	19.6	41.7	38.8	32.3	40.5	27.2
Days to 50% silking	33.7	27.1	39.2	29.6	30.2	40.1	22.9	42.1	35.0	38.0	39.1	22.9
Anthesis-silking interval	31.6	15.0	53.4	19.0	20.6	60.4	9.8	22.0	68.2	26.5	12.0	61.5
Plant height	41.2	35.6	23.2	26.1	23.6	50.3	32.5	42.4	25.1	32.3	31.5	36.3
>Ear height	24.9	27.1	48.0	33.3	28.8	37.9	29.3	32.0	38.7	34.0	37.4	28.6
Plant aspect	30.1	26.8	43.1	18.2	36.8	45.0	21.0	33.5	45.5	18.4	37.8	43.8
Ear aspect	22.0	36.2	41.8	19.4	18.8	61.9	22.9	22.6	54.5	17.8	24.1	58.1
Ears per plant	28.3	25.8	45.9	13.9	18.4	67.7	21.8	25.8	52.4	20.1	30.2	49.7
Stay-green characteristics	23.0	27.3	49.6	23.2	23.9	52.9	-	-	-	-	-	-
Tassel blast	-	-	-	21.8	12.1	66.1	-	-	-	-	-	-

**Table 4 plants-10-02596-t004:** GCA effects of the 24 early QPM inbred lines for grain yield and other agronomic traits evaluated under CDHS, low soil nitrogen, optimal and across test environments.

	Low Soil Nitrogen					Combined Drought and Heat Stress Conditions					Optimal Conditions		Across Test Environments	
	Grain Yield		Stay Green Characteristic		Grain Yield		ASI		Stay Green Characteristic		Grain Yield		Grain Yield	
**INBREDS**	GCAm	GCAf	GCAm	GCAf	GCAm	GCAf	GCAm	GCAf	GCAm	GCAf	GCAm	GCAf	GCAm	GCAf
TZEQI 106	310.57 *	174.91	−0.45 **	−0.40 *	−53.52	63.06	0.1	0.27	0.21	0.32 *	−179.3	−210.65	3.22	372.64
TZEQI 113	415.18 **	−10.83	−0.1	0.13	502.79 *	−265.53	−0.38 **	−0.05	−0.11	−0.31 *	266.62	−49.47	372.64	−96.19
TZEQI 122	−216.37	163.56	0.3	0.29	−521.41 *	−159.89	0.21	−0.13	0.03	−0.11	200.59	12.34	−116.3	26.39
TZEQI 123	−509.38 **	−327.64	0.26	−0.02	72.14	362.37	0.07	−0.09	−0.13	0.1	−287.91	247.78	−259.56	107.96
TZEQI 130	175.84	33.17	−0.32 *	0.01	133.17	96.02	−0.46 **	−0.21	0.1	0.03	−114.56	−128.97	52.9	−40.64
TZEQI 132	−91.59	−486.44 *	0.26	0.03	16.35	75.92	0.11	0.2	0.07	−0.34 *	192.46	189.82	3.88	−27.77
TZEQI 140	208.01	298.45	−0.45 **	0.11	−126.57	197.95	0	0.14	−0.44 **	0.43 *	143.61	9.55	111.06	138.82
TZEQI 143	−292.25 *	154.82	0.51 **	−0.15	−22.95	−369.89	0.35 *	−0.13	0.27 *	−0.13	−221.52	−70.39	−167.84	−70.42
TZEQI 158	567.43 **	353.41	−0.11	−0.21	242.39	17.36	0	0.01	−0.39 **	−0.04	25.9	−81.61	222.11	57.76
TZEQI 159	−94.48	−260.67	0.15	−0.07	−163.59	−390.22	−0.04	0.25	0.57 **	0.43 *	−259.52	−267.4	−109.65	−281.54
TZEQI 162	−154.07	−160	0.07	0	−19.1	233.1	−0.25	−0.13	−0.05	−0.14	−170.91	−83.24	−151.5	−50.1
TZEQI 165	−318.88 *	67.26	−0.12	0.28	−59.7	139.75	0.29 *	−0.13	−0.13	−0.25	404.53	432.25	39.05	273.87
TZEQI 171	−85.12	179.5	0.16	0.08	−52.95	−64.34	−0.34 *	−0.57 **	−0.12	−0.30 *	−59	−443.47	−57.96	−79.77
TZEQI 175	−201.68	−45.77	0.02	−0.07	249.93	266.38	0.30 **	−0.14	−0.17	−0.1	−346	254.71	−126.97	138.69
TZEQI 176	−85.69	−254.56	−0.19	0.13	−60.98	206.04	0.28 *	0.48 **	−0.12	0.35 *	87.82	29.46	−16.28	−17.64
TZEQI 188	372.49 *	120.83	0.01	−0.15	−136	−408.08	−0.25 *	0.23	0.42 **	0.05	317.17	159.3	201.2	−41.28
TZEQI 210	208.06	961.61 **	−0.01	−0.45 *	81.41	256.99	−0.08	−0.27 *	−0.25	0.50 **	−972.07	267.01	−382.49	452.05 *
TZEQI 216	−292.18 *	−575.11 *	−0.30 *	0.07	503.03 *	356.13	−0.05	−0.09	0.07	−0.12	−391.84	−61.44	−166.81	−104.27
TZEQI 219	−144.21	−101.23	0.02	−0.07	−633.01 **	−470.69	0.12	0.29 *	0.03	−0.52 **	1868.26 **	148.87	685.06 **	−74.52
TZEQI 228	228.33	−285.26	0.28 *	0.45 *	48.56	−142.44	0.01	0.07	0.15	0.14	−504.36	−354.45	−135.76	−273.26
TZEQI 240	77.6	−612.63 *	0.32 *	0.22	−185.6	−233.38	0.01	0.27	−0.13	−0.02	167.76	−716.16	42.49	−530.95 *
TZEQI 241	−312.11 *	60.19	0.13	0.04	−372.45	104.62	−0.12	−0.33 *	−0.28 *	0.12	−409.54	763.95	−388.39	410.10 *
TZEQI 246	−252.43	−121.31	0.16	0.02	230.7	−22.51	0.15	0.38 *	−0.18	0.29	−247.34	−225.03	−115.52	−136.02
TZEQI 6	486.95 **	673.75 **	−0.61 **	−0.28	327.35	151.28	−0.04	−0.32 *	0.59 **	−0.39 *	489.12	177.24	461.42 *	256.87
S.E	135.26	225.71	0.14	0.17	206.28	317.59	0.15	0.16	0.13	0.15	427.46	459.27	230.05	195.79

*, ** Significant at 0.05 and 0.01 probability levels.

**Table 5 plants-10-02596-t005:** Grain yield performance of selected QPM hybrids across the stress environments.

	Grain Yield				Yield Reduction (%)		
Hybrids	CDHS	Low-Nitrogen	Optimal	Across	Low-Nitrogen	CDHS	MI
TZEQI 6 × TZEQI 228	2134	3826	6386	4031	40.1	66.6	11.1
TZEQI 210 × TZEQI 188	1954	5388	6692	4582	19.5	70.8	11.0
TZEQI 6 × TZEQI 55(check)	2104	3832	6622	4716	42.1	68.2	9.4
TZEQI 113 × TZEQI 6	1613	4451	5643	3760	21.1	71.4	8.5
TZEQI 6 × TZEQI 210	2471	4483	6671	3613	32.8	63.0	8.4
TZEQI 241 × TZEQI 228	2699	4700	5145	3815	8.6	47.5	7.9
TZEQI 246 × TZEQI 210	2146	4255	5055	3720	15.8	57.6	7.1
TZEQI 241 × TZEQI 216	3604	3066	3807	3566	19.5	5.3	6.8
TZEQI 6 × TZEQI 219	1350	4314	6359	4607	32.2	78.8	6.6
TZEQI 39 × TZEQI 44(check)	3885	3877	6639	4627	41.6	41.5	5.6
TZEQI 39 × TZEQI 14(check)	2303	4435	5543	3854	20.0	58.5	5.6
TZEQI 171 × TZEQI 158	2206	3878	4784	3288	19.0	53.9	5.5
TZEQI 106 × TZEQI 6	1620	3307	3943	2858	16.1	58.9	4.9
TZEQI 210 × TZEQI 171	2344	3390	6506	3871	47.9	64.0	4.5
TZEQI 140 × TZEQI 113	2139	4227	5098	3384	17.1	58.0	4.2
TZEQI 113 × TZEQI 240	632	2475	4877	2626	49.3	87.0	-5.7
TZEQI 140 × TZEQI 106	664	2628	5161	2760	49.1	87.1	-6.4
TZEQI 165 × TZEQI 132	1804	2698	5391	2827	49.9	66.5	-6.4
TZEQI 130 × TZEQI 123	1418	2242	4780	2368	53.1	70.3	-6.6
TZEQI 132 × TZEQI 113	1236	2180	4990	2584	56.3	75.2	-7.0
TZEQI 175 × TZEQI 159	1124	1930	4916	2491	60.8	77.1	-7.2
TZEQI 132 × TZEQI 122	703	1511	5741	2548	73.7	87.8	-7.6
TZEQI 143 × TZEQI 122	342	2759	5037	2449	45.2	93.2	-7.7
TZEQI 159 × TZEQI 143	1045	2201	4883	2550	54.9	78.6	-9.9
TZEQI 159 × TZEQI 132	394	2003	4903	2368	59.1	92.0	^−1^2.8
Mean	1578	3057	5164	3158	44.9	67.0	
S.E	67	72	76	62			

**Table 6 plants-10-02596-t006:** List of the 24 early QPM inbred lines used in the North Carolina design II mating design to generate the 96 single-cross hybrids.

S/N	INBREDS	PEDIGREE	Reaction to Low-Nitrogen	Reaction to CDHS	SET
1	TZEQI 106	(TZEEQI 7 × TZEQI 6)F1 4/13 BC1 S7 1/2^−1^/1-3/4-3/3-2/2^−1^/1^−1^/1	S	S	A
2	TZEQI 113	(TZEEQI 7 × TZEQI 6)F1 4/13 BC1 S7 2/2-2/3^−1^/4-5/5-2/8^−1^/1^−1^/1	S	T	A
3	TZEQI 122	(TZEEQI 7 × TZEQI 6)F1 4/13 BC1 S7 2/2-2/3-4/4-6/6-3/4^−1^/2^−1^/1	T	T	A
4	TZEQI 123	(TZEEQI 7 × TZEQI 6)F1 4/13 BC1 S7 2/2-2/3-4/4-6/6-4/4^−1^/1^−1^/1	T	S	A
5	TZEQI 130	(TZEEQI 7 × TZEQI 6)F1 10/13 BC1 S7 1/3^−1^/1-3/5-2/3-3/3^−1^/1^−1^/1	T	T	B
6	TZEQI 132	(TZEEQI 7 × TZEQI 6)F1 10/13 BC1 S7 2/3^−1^/1-3/5^−1^/2^−1^/2^−1^/1^−1^/1	S	S	B
7	TZEQI 140	(TZEEQI 7 × TZEQI 6)F1 12/13 BC1 S7 1/2^−1^/1^−1^/3^−1^/4^−1^/2^−1^/1^−1^/1	T	S	B
8	TZEQI 143	(TZEEQI 7 × TZEQI 6)F1 13/13 BC1 S7 1/2^−1^/1^−1^/3-3/3-2/3^−1^/1^−1^/1	S	S	B
9	TZEQI 158	(TZEEQI 7 × TZEQI 4)F1 2/10 BC1 S7 1/2^−1^/3-2/3-2/3-2/3^−1^/1^−1^/1	T	S	C
10	TZEQI 159	(TZEEQI 7 × TZEQI 4)F1 2/10 BC1 S7 1/2^−1^/3-2/3-2/3-3/3^−1^/1^−1^/1	T	T	C
11	TZEQI 162	(TZEEQI 7 × TZEQI 4)F1 2/10 BC1 S7 1/2-3/3^−1^/4-5/5^−1^/2^−1^/1^−1^/1	T	T	C
12	TZEQI 165	(TZEEQI 7 × TZEQI 4)F1 2/10 BC1 S7 1/2-3/3-3/4-3/4-2/4^−1^/1^−1^/1	S	S	C
13	TZEQI 171	(TZEEQI 7 × TZEQI 4)F1 3/10 BC1 S7 1/2^−1^/2^−1^/1-2/2^−1^/3^−1^/1^−1^/1	S	T	D
14	TZEQI 175	(TZEEQI 7 × TZEQI 4)F1 3/10 BC1 S7 2/2-2/3^−1^/1^−1^/4^−1^/3^−1^/2^−1^/1	S	T	D
15	TZEQI 176	(TZEEQI 7 × TZEQI 4)F1 3/10 BC1 S7 2/2-2/3^−1^/1^−1^/4-2/3^−1^/2^−1^/1	T	S	D
16	TZEQI 188	(TZEEQI 7 × TZEQI 4)F1 3/10 BC1 S7 3/3^−1^/4-5/6^−1^/3-2/2^−1^/1^−1^/1	T	S	D
17	TZEQI 210	(TZEEQI 102 × TZEQI 6)F1 2/11 BC1 S7 2/2^−1^/1^−1^/4^−1^/3^−1^/2^−1^/1^−1^/1	S	T	E
18	TZEQI 216	(TZEEQI 102 × TZEQI 6)F1 9/11 BC1 S7 1/3-2/2^−1^/2-2/2-2/2^−1^/1^−1^/1	S	T	E
19	TZEQI 219	(TZEEQI 7 × TZEQI 60)F1 2/17 BC1 S7 2/2^−1^/1-2/3^−1^/1^−1^/3^−1^/1^−1^/1	S	T	E
20	TZEQI 228	(TZEEQI 137 × TZEQI 49)F1 2/9 BC1 S7 2/2^−1^/2^−1^/3-2/2-2/2^−1^/1^−1^/1	S	T	E
21	TZEQI 240	(TZEEQI 7 × TZEQI 6)F1 4/13 BC1 S7 1/2^−1^/1^−1^/2-4/4-2/2^−1^/1^−1^/1	T	S	F
22	TZEQI 241	(TZEEQI 7 × TZEQI 6)F1 4/13 BC1 S7 2/2-2/3-2/4^−1^/6-2/3^−1^/1^−1^/1	T	T	F
23	TZEQI 246	(TZEEQI 7 × TZEQI 4)F1 3/10 BC1 S7 3/3^−1^/4-3/6-2/3^−1^/3^−1^/1^−1^/1	T	S	F
24	TZEQI 6	CHECK	S	S	F

## Data Availability

The datasets used in the present study have been deposited at the IITA CKAN repository.

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
