# Peer review of "Genetic Analysis of Early White Quality Protein Maize Inbreds and Derived Hybrids under Low-Nitrogen and Combined Drought and Heat Stress Environments"

_plants, 2021, doi:10.3390/plants10122596_

Round 1
Reviewer 1 Report
The authors were attempted to address the major concerns of the global maize farming community, though it was planned for some parts of Africa. The work was planned well and executed. However, I would see careful checking of the manuscript by authors will improve the presentation. In some places, there were different fonts and many places had repeated sentences. All the best for the publication
Author Response
RESPONSE TO COMMENT OF REVIEWER ONE
The authors were attempted to address the major concerns of the global maize farming community, though it was planned for some parts of Africa. The work was planned well and executed. However, I would see careful checking of the manuscript by authors will improve the presentation. In some places, there were different fonts and many places had repeated sentences. All the best for the publication
Response
Thanks for the valuable comments. We sincerely appreciate the time and effort spent reviewing the manuscript. The differences in font size have been corrected according to the manuscript template and repeated sentences have been removed.

Reviewer 2 Report
Comments to the Author
The manuscript ‘Genetic Analysis of Early White Quality Protein Maize Inbreds Under Combined Heat and Drought Stressand Low-Nitrogen Environments’. The authors studied the traits of the early QPM inbred lines under CDHS, low-nitrogen, and optimal environments. The authors also assessed the performance of the inbreds in hybrid combinations. English language should be improved, I suggest asking language and style correction.
Title
Genetic Analysis of Early White Quality Protein Maize Inbreds Under Combined Heat and Drought Stressand Low-Nitrogen Environments.
Please add (in) before Maize and put space between stress and words.
Abstract
Page 1
Please summarize the aim of the study on one sentence because it is written in details the introduction.
Please write a good conclusion at the end of the abstract.
Introduction
The introduction is very long with several repeated sentences. Please improve the introduction.
Please check wheat production and update the reference because this reference year is 2017. The world’s production now is more than 1.05 billion tons.
You wrote (increase 12% from 193 million tons to 1.4 billion tons), I think these numbers are incorrect how come 12 % move from 193 million to 1.4 billion!
Page 2 please add to before 80%.
Please change to non-availability of fertilizers.
You wrote maize production and productivity, please delete productivity.
Please add reference to the sentence (A major challenge of maize breeders of the present generation is to develop maize cultivars with CDHS and low-nitrogen tolerance for the agro-ecological zones of SSA).
The introduction is not well organized.
Results
Please move each table and figure to its suitable place after the citation in the text directly.
Please avoid repeating the same meaning in different sentences every time.
Discussion
Please give your explanation of the results and don’t just say these results agree with or contrast with other studies. After that, you can compare this result with other studies’ results. The discussion looks like repeating the results and comparing with other studies’ results. Please rewrite the discussion again in a more scientific way.
Several grammatical errors were reported, please check the text carefully for grammatical errors.
Materials and methods
Please correct superphosphate (P2O5).
Abbreviations should be explained for the first time in the text.
Could you confirm your results through molecular biology techniques?
Author Response
RESPONSES TO COMMENTS OF REVIEWER TWO
The manuscript ‘Genetic Analysis of Early White Quality Protein Maize Inbreds Under Combined Heat and Drought Stressand Low-Nitrogen Environments’. The authors studied the traits of the early QPM inbred lines under CDHS, low-nitrogen, and optimal environments. The authors also assessed the performance of the inbreds in hybrid combinations. English language should be improved, I suggest asking language and style correction.
Response
Thanks for the accurate summary and the invaluable comments. The English language has been revised as recommended.
- Title: Genetic Analysis of Early White Quality Protein Maize Inbreds Under Combined Heat and Drought Stressand Low-Nitrogen Environments. Please add (in) before Maize and put space between stress and words.
Response
A space has been inserted between the words “stress” and “and” (Line 3, Page 1).
Please note that it will not be appropriate to include “in” before “maize”.
- Abstract: Please summarize the aim of the study on one sentence because it is written in details the introduction.
Response
The objectives of the study have been revised into a sentence as requested.
2.1 Please write a good conclusion at the end of the abstract.
Response
The abstract has been revised.
- Introduction: The introduction is very long with several repeated sentences. Please improve the introduction.
Response
The introduction has been revised and sentences that were similar have been removed.
3.1. Please check wheat production and update the reference because this reference year is 2017. The world’s production now is more than 1.05 billion tons. You wrote (increase 12% from 193 million tons to 1.4 billion tons), I think these numbers are incorrect how come 12 % move from 193 million to 1.4 billion!
Response
The statistics for the global maize production for 2019/2020 has been revised as suggested and included in the manuscript (Line 35, Page 1).
3.2 Page 2 please add “to” before 80%.
Response
The word “to” has been included in Line 50, Page 2 as requested.
3.3 Please change to non-availability of fertilizers.
Response
It has been changed to “non-availability of fertilizers” with the inclusion of letter “s” (Line 52, Page 2) as requested.
3.4 You wrote maize production and productivity, please delete productivity.
Response
The word “productivity” has been deleted (Line 54, Page 2).
3.5 Please add reference to the sentence (A major challenge of maize breeders of the present generation is to develop maize cultivars with CDHS and low-nitrogen tolerance for the agro-ecological zones of SSA).
Response
A relevant reference has been added (Line 66, Page 2).
3.5 The introduction is not well organized.
Response
The introduction has been improved.
4.0 Results: Please move each table and figure to its suitable place after the citation in the text directly.
Response
Tables and Figures have been placed after the citations.
4.1 Please avoid repeating the same meaning in different sentences every time.
Response
The repetitions have been removed throughout the manuscript as suggested.
5.0 Discussion: Please give your explanation of the results and don’t just say these results agree with or contrast with other studies. After that, you can compare this result with other studies’ results. The discussion looks like repeating the results and comparing with other studies’ results. Please rewrite the discussion again in a more scientific way. Several grammatical errors were reported, please check the text carefully for grammatical errors.
Response
The discussion section has been revised as suggested.
6.0 Materials and methods: Please correct superphosphate (P2O5).
Response
The superphosphate designation has been changed to P2O5 as requested.
6.1 Abbreviations should be explained for the first time in the text.
Response
The abbreviations have been explained for the first time they are used.
6.2 Could you confirm your results through molecular biology techniques?
Response
Thank you for the recommendation. However, this is beyond the scope of our study.

Reviewer 3 Report
Dear Authors,
Reviewer comments plants-1453433
The manuscript entitled „Genetic analysis of early white quality protein in maize inbreds under combined heat and drought stress and low-nitrogen environments“ represents a useful study aimed at an investigation of grain yield components and other agromorphological characteristics in a set of maize inbred lines tested at two locations, Mokwa and Kadawa, characterized by drought and low-nitrogen conditions in Nigeria.
I have only a few minor comments on the present manuscript:
Materials and methods, locations characteristics: In Materials and methods, supplementary Figure S3, I think that more detailed data should be provided on the growth temperature (not just average monthly temperature, but, e.g., daily maximum and minimum temperature, and also rainfall although it was scarce, during the whole plant growing season in both years 2019 and 2020).
Formal comments on the text:
Title: Add the word „in“ between the words „protein“ and „maize“ and add a space between the words „stress“ and „and“ in the manuscript title „Genetic analysis of early white quality protein in maize inbreds under combined heat and drought stress and low-nitrogen environments“
Introduction, the first line: Add a space in the plant scientific name „Zea mays“.
Introduction, page 2, line 5: Add the word „with“ following the verb „fed“ in the statement „…on infants fed with QPM porridge…“
Introduction, page 2, line 25: Correct the typing error in the word „germplasm“ (not „germpalsm“).
Introduction, page 2, line 25: Add „a“ preceding the word „lack“ in the statement „a lack of tolerance to low-nitorgen…“
Figure 5 legend: Correct the typing error in the term „ASI = anthesis-silking interval“ (not „antheisi-silking interval“).
Discussion, page 8, lines 32, 34: Add the word „ones“ in the statements „…that non-additive gene effects were more important than the additive ones“ ; „non-additive gene effects over the additive ones“.
Materials and methods: use reference numbers instead of publication years in the references „Badu-Apraku et al., 2016“, „Badu-Apraku and Fakorede, 2017“.
Final recommendation: Accept after a minor revision.

Author Response
RESPONSES TO COMMENTS OF REVIEWER THREE
The manuscript entitled „Genetic analysis of early white quality protein in maize inbreds under combined heat and drought stress and low-nitrogen environments“ represents a useful study aimed at an investigation of grain yield components and other agromorphological characteristics in a set of maize inbred lines tested at two locations, Mokwa and Kadawa, characterized by drought and low-nitrogen conditions in Nigeria. I have only a few minor comments on the present manuscript:
Response
We sincerely appreciate the time and effort spent reviewing the manuscript and the invaluable contributions.
- Materials and methods, locations characteristics: In Materials and methods, supplementary Figure S3, I think that more detailed data should be provided on the growth temperature (not just average monthly temperature, but, e.g., daily maximum and minimum temperature, and also rainfall although it was scarce, during the whole plant growing season in both years 2019 and 2020).
Response
The coordinates of the locations and characteristics have been provided in the materials and methods section. Also, the daily maximum and minimum temperatures for the combined drought and heat trial sites for 2020 and 2021 have now been included as Supplementary Tables 2 and 3.
- Formal comments on the text:
Title: Add the word „in“ between the words „protein“ and „maize“ and add a space between the words „stress“ and „and“ in the manuscript title „Genetic analysis of early white quality protein in maize inbreds under combined heat and drought stress and low-nitrogen environments“.
Response
A space has been inserted between the words “stress” and “and” (Line 3, Page 1) as requested.
Please note that it will not be appropriate to include “in” before “maize”.
2.1 Introduction, the first line: Add a space in the plant scientific name „Zea mays“.
Response
A space has been inserted between the words “Zea” and “mays” (Line 35, Page 1) as requested.
2.1 Introduction, page 2, line 5: Add the word „with“ following the verb „fed“ in the statement „…on infants fed with QPM porridge…“
Response
The word “with” has been inserted after “fed” and “and” (Line 44, Page 2) as requested.
2.2 Introduction, page 2, line 25: Correct the typing error in the word “germplasm“ (not “germpalsm“).
Response
The correction has been made (Line 65, Page 2) as requested.
2.3 Add “a“ preceding the word “lack“ in the statement “a lack of tolerance to low-nitorgen…“.
Response
Please, it will not be appropriate for “a” to precede “lack” as requested.
2.4 Figure 5 legend: Correct the typing error in the term „ASI = anthesis-silking interval“ (not „antheisi-silking interval“).
Response
The corrections has been made (Line 65, Page 2) as requested.
- Discussion, page 8, lines 32, 34: Add the word „ones“ in the statements „…that non-additive gene effects were more important than the additive ones“ ; „non-additive gene effects over the additive ones“.
Response
Please it will not be appropriate to add “ones” in the statements as requested.
4.0 Materials and methods: use reference numbers instead of publication years in the references „Badu-Apraku et al., 2016“, „Badu-Apraku and Fakorede, 2017“.
Response
The appropriate reference number has been included.
5.0 Final recommendation: Accept after a minor revision.
Response
Thanks for your invaluable comments and suggestion.

Round 2
Reviewer 2 Report
The manuscript ‘Genetic Analysis of Early White Quality Protein Maize Inbreds Under Combined Heat and Drought Stress and Low-Nitrogen Environments’.
Please rephrase the following sentences to be more understandable:
With the current global yield of 1.1 billion tons, maize production ......etc.
The heterotic groups identified by the HGCAMT method across environments would increase the chances of developing novel and outstanding early maturing QPM hybrids and synthetics with CDHS.... etc.
Please check the whole text for minor grammatical errors.
Author Response
RESPONSES TO COMMENTS OF REVIEWER TWO
Please rephrase the following sentences to be more understandable:
With the current global yield of 1.1 billion tons, maize production ......etc.
Response
The sentence has been rephrased as requested.
The heterotic groups identified by the HGCAMT method across environments would increase the chances of developing novel and outstanding early maturing QPM hybrids and synthetics with CDHS.... etc.
Response
The sentence has been rephrased as requested.
Please check the whole text for minor grammatical errors.
Response
The whole text has been checked for minor grammatical errors as requested.
